# Factors Influencing the Willingness of Palliative Care Utilization among the Older Population with Active Cancers: A Case Study in Mandalay, Myanmar

**DOI:** 10.3390/ijerph18157887

**Published:** 2021-07-26

**Authors:** Aye Tinzar Myint, Sariyamon Tiraphat, Isareethika Jayasvasti, Seo Ah Hong, Vijj Kasemsup

**Affiliations:** 1ASEAN Institute for Health Development, Mahidol University, Salaya, Phutthamonthon, Nakhon Pathom 73170, Thailand; ayetzmyint.cbl@gmail.com (A.T.M.); seoah.hon@mahidol.ac.th (S.A.H.); vijj.kas@mahidol.ac.th (V.K.); 2Institute of Nutrition, Mahidol University, Salaya, Phutthamonthon, Nakhon Pathom 73170, Thailand; graphkodomo@gmail.com; 3Faculty of Medicine Ramathibodi Hospital, Mahidol University, Bangkok 10400, Thailand

**Keywords:** palliative care, hospice, cancer patients, older populations

## Abstract

Palliative care is an effective, multidisciplinary healthcare service to alleviate severe illness patients from physical, psychological, and spiritual pain. However, global palliative care has been underutilized, especially in developing countries. This cross-sectional survey aimed to examine the factors associated with older cancer patients’ willingness to utilize palliative care services in Myanmar. The final sample was composed of 141 older adults, 50-years of age and above who suffered from cancers at any stage. Simple random sampling was applied to choose the participants by purposively selecting three oncology clinics with daycare chemotherapy centers in Mandalay. We collected data using structured questionnaires composed of five sections. The sections include the participant’s socio-economic information, disease status, knowledge of palliative care, psychosocial and spiritual need, practical need, and willingness to utilize palliative care services. The study found that approximately 85% of older cancer patients are willing to receive palliative care services. The significant predictors of willingness to utilize palliative care services include place of living, better palliative care knowledge, more need for spiritual and psychosocial support, and practical support. This study can guide health policymakers in increasing the rate of palliative care utilization. The suggested policies include developing community-level palliative care services in Myanmar, especially in rural areas, promoting palliative care knowledge, applying appropriate religious and spiritual traditions at palliative treatment, and developing suitable medicines for the critically ill.

## 1. Introduction

Palliative care is a multidisciplinary healthcare process to improve patients’ quality of life and their families by alleviating or reducing suffering from severe illness patients’ physical, psychological, and spiritual pain [1]. In 1967, British physician Cicely Saunders, the widely accepted founder of modern palliative care, built St. Christopher’s Hospice to provide specialist care to the incurably ill, located in South East London [2,3]. Saunders’ early work is the fundamental concept of palliative care, recognizing that pain is a physiological experience and encompasses spiritual, psychological, and social dimensions [3]. In the early 1980s, the World Health Organization (WHO) officially began developing a global palliative initiative to advocate for pain relief, emphasizing cancer pain relief [1] and updated the term of palliative care in 2002 as “an approach that improves the quality of life of patients and their families facing the problem associated with life-threatening illness, through the prevention and relief of suffering by means of early identification and impeccable assessment and treatment of pain and other problems, physical, psychosocial, and spiritual” [4,5].

Up to now, there has been a rapid rise in demand and supply of palliative care among Western countries, including England, European countries, Canada, and the United States, comprising home palliative care [2] as well as in- and out-patient departments [2,6]. Among the Eastern countries, South Korea is the well-known pioneer for delivering hospice palliative care, focusing on terminal cancer patients [7]. Moreover, Japan, Australia, and New Zealand have well-established pain management services and training programs for palliative care [8]. For South-East Asia countries, palliative care has not been widespread [8]. In the mid-1980s, Singapore and the Philippines first developed palliative care. Later, in the 1990s, Malaysia, Thailand, and Indonesia began to utilize the palliative care concept. However, other countries such as Brunei, Cambodia, Myanmar, Vietnam, and Laos have only been developing it over the past 10 years [8,9].

Utilizing palliative care can strengthen the quality of life and improve efficacy in controlling physical symptoms and psychosocial and spiritual concerns, including improved pain and symptom control, hospital admissions or stays, and anxiety or depression among patients with severe illness [10]. Palliative care can diminish several disease burdens, meet caregivers’ overall satisfaction, and even extend the survival rate of advanced cancer stage patients [1,11]. As recommended by WHO, palliative care is applicable early in the course of illness, in conjunction with other therapies, such as chemotherapy or radiation therapy, to engender patients with a prolonged life span [5,12]. However, palliative care has been offered only at very advanced stages, a short time before dying [12]. Most of the population in the world was still unaware of this palliative care which consequently results in underutilization. As stated by WHO, out of 232 countries globally, only 20 countries (8.6%) had integrated palliative care as part of their health care system [13]. Moreover, 80% of low- and middle-income countries had a severe shortage of palliative care facilities [14]. Previous research indicated that critical barriers to utilization of palliative care might include lack of adequate education/training and perception of palliative care, inadequate size of palliative medicine-trained workforce, and lack of sufficient reimbursement for palliative care and regulatory barriers [15,16].

The patients’ voice in developing, improving, and measuring service quality to increase palliative utilization is challenging [17]. However, research on palliative care at the patient level is complicated and limited [12]. Thus, it is essential to increase more palliative studies from patients’ perspectives. Serious illness can cause many different types of suffering, whether physical, emotional or spiritual. Unexpectedly, more than 80% of patients with advanced, incurable cancer still have unmet needs, including those with symptoms and psychological conditions [18,19]. A few researchers investigated patients’ willingness to utilize palliative care services [12,17,19]. They found that emotional/spiritual support [19], such as treatment of anxiety, depression, weakness, and the facilitation of religious activities and services [17], were associated with an interest in receiving palliative treatment [12,17,19]. Moreover, unmet physical needs such as pain [20,21,22], impairing autonomy and mobility, including weakness, tiredness or needing assistance with daily living activities [23] influenced patients’ preference in palliative treatment. In addition to unmet needs, knowledge about palliative care [24,25,26,27] is a significant positive factor in the willingness to use palliative medication. A previous study indicated that patients had a greater willingness to use palliative care in the early treatment after receiving education on palliative care [28]. Other associated factors that potentially influence patients’ readiness to use palliative care services include gender, education level, age, race, disease aggressiveness, financial strain [29,30], religion [31], and marital status [32]. Regarding social support, there was a strong relationship between the support of primary caregivers and the use of palliative care in cancer patients [31], and patients in rural areas are at a lower utilization rate than those in urban areas [32].

Myanmar, a low-and middle-income country in Southeast Asia, faces insufficient health care services [33], including palliative care delivery [8]. There are only two palliative care units in Myanmar, Yangon General Hospital and Mandalay General Hospital [34]. To increase the utilization rate, we need to understand what factors are associated with patients’ willingness to utilize palliative care services. This cross-sectional survey aimed to examine the factors that might correlate with cancer patients’ willingness to utilize palliative care services in the Myanmar population. The study results can provide helpful knowledge to guide health policymakers in meeting future needs, as well as improving the quality of palliative care services to increase the utilization rate.

## 2. Methodology

### 2.1. Description of Survey and Study Population

This survey is a cross-sectional study design in a quantitative method. The study was conducted at three private oncology clinics in Mandalay, Myanmar, while the COVID-19 pandemic occurred by following the social-distancing rule and wearing personal protective equipment.

We performed the pilot study at one private clinic with 30 adults, 50-years of age and above who suffered from any cancer stage and met the inclusion criteria. From the pretest, the prevalence of willingness to utilize palliative care services was 85% (0.85). We calculated the sample size using the standard deviation. In addition, we assumed that the estimated proportion of willingness to utilize palliative care services among cancer patients is equal to 0.85. With a 5% margin of error (ME) and a z-score (z) of 1.96, our formula for the sample size translates from: Sample size = (z^2^ × (p (1 − p)))/ME^2^ to sample size = (1.96^2^ × (0.85(1 − 0.85)))/0.05^2^. = 196. Simple random sampling was applied to choose 196 participants by purposively selecting the three oncology clinics with daycare chemotherapy centers in Mandalay. Due to the difficulty in data collection as a result of the COVID pandemic, our final sample size between 1–30 September 2020 was 141, with a response rate of approximately 75%. This research project received ethical approval from the “Research Ethics Committee of the Faculty of Social Sciences and Humanities, Mahidol University” (Certificate of Approval No. 2020/059.2503). Informed consent was obtained from all study participants.

### 2.2. Measures

The research tool was composed of structured questionnaires including socio-demographic characteristics, medical history, knowledge, and need for palliative care concerning the willingness to utilize palliative care services. The questionnaires consisted of five domain parts, and we conducted face-to-face interviews to collect the data. Part I: Socio-demographic characteristics include the age of the respondent in the completed years, gender of the respondents (male or female), the religion of the respondents (Buddhist, Christian, Muslim, Hinduism), marital status, and spousal characteristics (single or cohabitating, married, separated, divorced). The education level of the respondents includes 0 for illiterate, 1 for reading and writing, 2 for primary level education, 3 for secondary level education, and 4 for tertiary level education. Family income is the total amount of money received by households and residence places, including urban and rural areas, recited by the respondents. We categorized family income using the percentile for the cut point: Low/medium (for a family with income below 75th percentiles) and high (with income at 75th percentiles or higher). Social networks include the source of support received by patients using the structural-functional social support scale of nine items (SFSS instrument) [35]. We summed up all the items and classified them into two levels, as low and high, using the mean cut point. Part II is made up of the medical history of patients, type of cancer, and stage of cancer (0 = stage 1 and 2 and 1 = stage 3 and 4). Part III is the knowledge of palliative care (PaCKS instrument), which comprises 13 items and Yes/No/Do Not know questions [36]. Part IV is the patient’s needs in relation to palliative care, including psychosocial and spiritual needs with nine items and practical needs with four items [37]. Spiritual and psychosocial needs were categorized as low and high using the mean for the cut point. Practical needs were also classified as low and high using the mean for the cut point. Part V is the willingness to use palliative care services? (Yes/No).

For reliability, the pilot test was carried out with 30 cancer patients in different study areas to make sure of the questions consistency. We analyzed Cronbach’s coefficient alpha to check the reliability and internal consistency. The Cronbach’s alpha for knowledge regarding palliative care was 0.82, the psychosocial and spiritual need was 0.81 items, and practical need was 0.67.

### 2.3. Data Analysis

Firstly, we examined the prevalence of willingness to use palliative care. Then, we analyzed the data with descriptive statistics. After that, we performed the Chi-square test to explore the association between the categorized factors and the willingness to use palliative care services. Later, we predicted the significant predictors of the willingness to use palliative care services using simple/multiple logistic regression by the Statistical Package for the Social Science (SPSS), 26th version. 

## 3. Results

### 3.1. Prevalence of Willingness to Use Palliative Care Services

The prevalence of willingness to use palliative care services among older cancer Myanmar patients was 85.8%, as shown in Table 1.

### 3.2. Percentage of Respondents in the Study

The total study sample included 141 Myanmar older adults (50 years or more) with active cancers. Almost three-fourths (71.6%) of the sample were women. More than half (53.9%) were 50 to 59 years old and were married (59.6%). Most had low education (44%), had an income not more than 700,000 Kyats/year (76.6%), and 53.9% lived in rural areas. More than half of the older adults (53.2%) had been cancer Stage 1/2, almost 40% of the respondents had breast cancer, and almost 60% lived with a low social network. From the total score of 13 for palliative knowledge, the average score of the sample was 10.19, with a standard deviation of 2.41 (see Table 2).

### 3.3. Percentage of Spiritual and Psychosocial Needs and Practical Need among the Patients

The mean score of spiritual and psychosocial needs was 34.7 with a standard deviation of 7.3, from the total score of 45. At the cut point 35, patients with low spiritual and psychosocial needs are 44.7%, whereas high psychological needs are 55.3%. Most patients (90.1%) need “I need spiritual support”, whereas the item that a small number of patients need (34.8%) is “I want to talk with someone about dying” (see Table 3). Regarding practical need, from the total score of 20, the mean score was 12.9, with a standard deviation of 4.95. At the cut point 13, patients with low practical needs are 51.1%, whereas high practical needs are 48.9%. Most patients (75.2%) need “I need help to manage my medication”, whereas the item that a small number of patients need (18.4%) is “I need help with my daily activities such as showering” (see Table 3).

### 3.4. Association between Categorized Independent Variables and Willingness to Utilize Palliative Care

The results of χ^2^ tests showed that significant factors associated with the willingness to utilize palliative care among Myanmar older adults are place of living, practical need, and spiritual and psychosocial needs. Namely, the patients living in rural areas were associated with more willingness to utilize the palliative treatment. Furthermore, practical needs and spiritual and psychosocial needs were significantly associated with the willingness among Myanmar older adults. Namely, a higher level of practical, spiritual, and psychosocial needs was associated with more willingness to utilize palliative care among Myanmar older patients with cancer (see Table 4).

### 3.5. Predictors of Willingness to Utilize Palliative Care

Table 5 indicates significant predictors of willingness to utilize palliative care among older adults. The results of multivariable logistic regression showed that the significant willingness predictors of using palliative care are place of living, palliative knowledge, need for practical support, and need for spiritual and psychosocial support. The older adults in rural areas were more likely to be willing to utilize palliative care than those in urban area by 3.62 times. The older adults with better palliative knowledge were more likely to be prepared for palliative utilization than those with insufficient palliative knowledge by 1.28 times. Furthermore, older adults who reported a higher need for practical support as well as spiritual and psychosocial support were more likely to be ready for palliative utilization than those with lower needs by 5.27 and 3.76 times, respectively. 

## 4. Discussion

This research indicated that approximately 85% of older cancer patients are willing to receive palliative care services. The significant predictors of willingness to utilize palliative care services include place of living, palliative care knowledge, the need for spiritual and psychosocial support, and practical support. Namely, the patients with more spiritual and psychosocial needs are more likely to be willing to use palliative care 3.76 times than those with a lower need for this support. Regarding practical support, we found that the patients with more need for practical support are more likely to be willing to use palliative care 5.27 times than those with a lower need for support. In addition, patients in rural areas are more likely to be ready to utilize palliative services 3.62 times than those in urban areas. In addition, patients with better knowledge of palliative care are more likely to use palliative care than those with lower palliative knowledge by 1.28 times. 

Regarding geographic living, previous research [32,38,39] consistently reported a higher rate of palliative care utilization in patients living in urban than in rural areas. The results may be possible in that people living in rural areas may encounter distance limitation to the facilities and difficulty accessing those services, contributing to the lower utilization rate [32]. However, our result is not the same as the previous research above. We found that patients living in rural areas were more willing to utilize palliative care services than patients in urban areas. Earlier research from Myanmar revealed that older adults in rural areas have unsafe socio-demographic conditions and low healthcare-seeking behavior [40]. Another study supported that Myanmar cancer patients with low socio-economic status (SES) had a significantly worse quality of life, perceived poor health care coordination, and were more likely to report severe pain than those with high SES [34]. Therefore, it is reasonable to consider that Myanmar patients living in rural areas may encounter a low quality of care and difficulty accessing health care appropriately. Thus, they may prefer to rely on an alternative medicine associated with traditional spiritual care as palliative practice. With this reason, we strongly believe that the developing community-level palliative care services in Myanmar will be able to facilitate people, especially in rural areas to have more quality of life and care.

For palliative knowledge, our results support the previous study addressing that knowledge about palliative care enhanced patients’ preference for palliative [25] and hospice utilization [24]. They found a strong positive relationship between hospice knowledge and favorable attitudes about hospice and that more favorable attitudes were associated with a preference for hospice [24]. This result may be possible in that palliative knowledge persons are more likely to be aware of the benefits of the services and thus be willing to afford these services. Therefore, it is challenging to develop more effective educational practices about palliative care for fortifying palliative knowledge and positive attitudes to raise palliative care utilization.

Concerning practical support, we found that patients with more need for practical support will be more willing to select palliative care for their treatment. Our result reaches an agreement with earlier studies [20,21,22], claiming that unmet physical needs among severe patients such as pain impair autonomy and mobility, including weakness, tiredness or needing assistance with daily living activities [23], influenced patients’ preference in palliative care. Interestingly, we found that the most need in practical support is the item “I need help to manage my medication,” with 75% of patients’ rating. Our finding supported the WHO message that one-third of the global population lacks reliable access to needed medicines, especially 50% of the people living in Africa and Asia countries [41]. Therefore, our results may imply that Myanmar’s population faces inadequate access to the essential drug, such as cancer medicine for a medical cure. Thus, they try to seek alternative medication for the best treatment. However, it is disappointing that the absence of palliative care, especially in developing countries, is associated with poorly managed pain and other symptoms and the lack of potent analgesics and other medications [42]. Therefore, we strongly agree that developing medicines for the critically ill and the dying is challenging and demands or requires political action in developing countries.

Beyond the practical dimension, our result indicated that the spiritual and psychosocial need is another significant predictor of the willingness to use palliative care services. Our finding was similar to that of previous research [12,17,19], supporting the association between attentiveness in receiving palliative care and the need for psychological support, such as treatment of anxiety, depression, weakness, and the facilitation of religious activities and services. Captivatingly, responses to the questionnaire section on the psychosocial and spiritual need, the item “I need spiritual support” received a 90% rating from patients. Spiritual is a sense of peace and determination inside our being, whereas religion is a construct of human making to express spirituality [43]. Across cultures, spirituality and religion continue to play an essential role in medical healing, especially when patients face the crisis of advanced illnesses towards the end of life [43]. A study in Southeast Asia [44], specifically Indonesia, supported spirituality/religious aspects as a palliative care provision. They found a significant element in the Indonesian palliative care service and many different religions among the participants. All the participants in palliative services reported being affiliated with particular religions and performing their religious practices daily with much comfort [44]. For Myanmar, spirituality plays a crucial role in patients’ treatment decisions, and faith-based spiritual techniques used by Buddha monks are standard in community services [45]. Thus, it is reasonable to find that most Myanmar patients need spiritual support to relieve the misery at the end of life. Therefore, understanding and applying religious and spiritual traditions is crucial to the patients’ holistic health in palliative treatment.

There are some limitations to this study. First, the nature of the study’s cross-sectional design cannot confirm the causal relationships between the willingness to utilize palliative care services and the predictors. Thus, future research may invite more cases and use structural equation modeling to explain the direct and indirect effect of the willingness. Second, since the target population in this study is only cancer patients, then we should add other severe illness patients to investigate the willingness to utilize palliative services in future studies. However, the strength of the present study is that it is the first study to investigate the willingness to use palliative care services among older adults in Myanmar, where the rate of utilization is quite low. Therefore, it is a valuable study that can notify policymakers to understand the patients’ perspectives in order to increase the utilization rate.

## 5. Conclusions

This cross-sectional survey examined the factors associated with cancer patients’ willingness to utilize palliative care services in the Myanmar population. The study found that approximately 85% of older cancer patients are willing to receive palliative care services. The significant positive predictors of willingness to utilize palliative care services include place of living, better palliative care knowledge, more need for spiritual and psychosocial support, and practical support. This study can guide health policymakers in growing the rate of palliative care utilization. The suggested policies include developing community-level palliative care services in Myanmar, especially in rural areas, promoting palliative care knowledge, applying appropriate religious and spiritual traditions at palliative treatment, and developing suitable medicines for the critically ill.

## Figures and Tables

**Table 1 ijerph-18-07887-t001:** Prevalence of willingness to use palliative care services.

Willingness to Use Palliative Care Services	Number	Percent
Yes	121	85.8
No	20	14.2

**Table 2 ijerph-18-07887-t002:** Percentage of the sample population.

Socio-Demographic Characteristics	Number (*n*)	Percentage (%)
**Age**		
50–59	76	53.9
60–69	42	29.8
70 year and higher	23	16.3
**Gender**		
Male	40	28.4
Female	101	71.6
**Marital Status**		
Married	84	59.6
Others (Single, Divorced, Separated, Widow)	57	40.4
**Education**		
Low Education (no school/primary school)	62	44
Medium Education (Middle/high school)	53	37.6
High Education (College and higher)	26	18.4
**Family Income** **(using the 75th percentile = 700,000 Kyats/year as the cut point)**		
Low/medium (0–700,000 Kyats/year)	108	76.6
High (more than 700,000 Kyats/year)	33	23.4
**Place of living**		
Rural	76	53.9
Urban	65	46.1
**Social Network** **(using the mean = 40 as the cut point)**		
Low (≤40)	84	59.6
High (>40)	57	40.4
**Stage of cancer**		
Stage 1/2	75	53.2
Stage 3/4/metastasis	66	46.8
**Type of cancer**		
Breast cancer	55	39.0
Genitourinary cancers	32	22.7
Respiratory tract cancers	21	14.9
Gastrointestinal cancers	31	22.0
Others	2	1.4
**Continuous variable**		
Knowledge of palliative care (Total score = 13)	{mean (SD) = 10.19 (2.41)}	

**Table 3 ijerph-18-07887-t003:** Percentage of spiritual and psychosocial needs and practical need among the patients.

I. Spiritual and Psychosocial Needs	No		Yes	
	Number	Percent	Number	Percent
I need emotional support.	29	20.6	112	79.4
I want to talk to someone about dying.	92	65.2	49	34.8
My family or friends need emotional support.	25	17.7	116	82.3
I need spiritual support.	14	9.9	127	90.1
I want someone to talk to my family or friends about my illness.	37	26.2	104	73.8
I want help to find meaning in my cancer experience.	38	27	103	73
I want to prepare now for what might happen in the future.	35	24.8	106	75.2
I want to talk to someone who understands what I am going through.	29	20.6	112	79.4
I want my family and friends to prepare now for what might happen in the future.	34	24.1	107	75.9
Score (Mean ± SD)	34.7 (7.3)			
Low spiritual and psychosocial needs (≤35)	63 (44.7%)			
High spiritual and psychosocial needs (>35)	78 (55.3%)			
**II. Practical Need**	**No**		**Yes**	
	**Number**	**Percent**	**Number**	**Percent**
My family or friends need help with my physical care.	58	41.1	83	58.9
I need help to manage physical symptoms such as pain.	39	27.7	102	72.3
I need help to manage my medication.	35	24.8	106	75.2
I need help with my daily activities such as showering.	115	81.6	26	18.4
Score (Mean + SD)	12.9 (4.05)			
Low practical needs (≤13)	72 (51.1%)			
High practical needs (>13)	69 (48.9%)			

**Table 4 ijerph-18-07887-t004:** Association between categorized independent variables and willingness to utilize palliative care.

Category Variables	Willingness to Utilize Palliative Care				Chi-Square	*p*-Value
	No		Yes			
	n	%	n	%		
**Age**						
50–59	11	14.50%	65	85.50%	0.38	0.83
60–69	5	11.90%	37	88.10%		
70 years and higher	4	17.40%	19	82.60%		
**Gender**						
Male	8	20.00%	32	80.00%	1.56	0.21
Female	12	11.90%	89	88.10%		
**Marital Status**						
Married	13	15.50%	71	84.50%	0.29	0.60
Others (Single, Divorced, Separated, Widow)	7	12.30%	50	87.70%		
**Education**						
Low Education (no school/primary school)	8	12.90%	54	87.10%	0.67	0.72
Medium Education (Middle/high school)	7	13.20%	46	86.80%		
High Education (College and higher)	5	19.20%	21	80.80%		
**Family Income** **(using the 75th percentile = 700,000 Kyats/year as the cut point)**						
Low/medium (0–700,000 Kyats/year)	16	14.80%	92	85.20%	0.15	0.70
High (more than 700,000 Kyats/year)	4	12.10%	29	87.90%		
**Place of living**						
Rural	6	7.90%	70	92.10%	5.36	0.02
Urban	14	21.50%	51	78.50%		
**Social Network** **(using the mean = 40 as the cut point)**						
Low (≤40)	14	16.70%	70	83.30%	1.05	0.31
High (>40)	6	10.50%	51	89.50%		
**Stage of cancer**						
Stage 1/2	9	12.00%	66	88.00%	0.63	0.43
Stage 3/4/metastasis	11	16.70%	55	83.30%		
**Cancer type**						
Breast cancer	6	10.9%	49	89.1%	0.80	0.37
Others	14	16.3%	72	83.7%		
**Spiritual and psychosocial needs** **(using the mean = 35 as the cut point)**						
Low (≤35)	15	23.80%	48	76.20%	8.67	0.00
High (>35)	5	6.40%	73	93.60%		
**Practical need** **(using the mean = 13 as the cut point)**						
Low (≤13)	16	22.20%	56	77.80%	7.81	0.01
High (>13)	4	5.80%	65	94.20%		

**Table 5 ijerph-18-07887-t005:** Simple and multiple logistic regression analysis to determine factors associated with willingness to utilize palliative care.

Variables	Crude OR (95% C.I.)	Adjusted OR (95% C.I.)	*p*-Value
**Age (years)**			
50–59	1.24 (0.36–4.36)	NS	
60–69	1.56 (0.37–6.49)	NS	
70 years and higher	1		
**Gender**			
Male	0.54 (0.20–1.44)	NS	
Female	1		
**Marital Status**			
Married	0.77 (0.29–2.05)	NS	
Others (Single, Divorced, Separated, Widow)	1		
**Education**			
Low Education (No school/primary school)	1.61 (0.47–5.48)	NS	
Medium Education (Middle/high school)	1.57 (0.45–5.51)	NS	
High Education (College and higher)	1		
**Family Income** **(using the 75th percentile = 700,000 Kyats/year as the cut point)**			
High (more than 700,000 Kyats/year)	1.26 (0.39–4.07)	NS	
Low/medium (0–700,000 Kyats/year)	1	1	
**Place of living**			
Rural	3.20 (1.15–8.90)	3.62 (1.06–12.33)	0.04 *
Urban	1		
**Social Network** **(using the mean = 40 as the cut point)**			
High (>40)	1.70 (0.61–4.73)	NS	
Low (≤40)	1		
**Stage of cancer**			
Stage 1/2	1.47 (0.57–3.80)	NS	
Stage 3/4/metastasis	1		
**Cancer type**			
Breast cancer	1.59 (0.57–4.42)	NS	
Others	1		
**Spiritual and psychosocial needs** **(using the mean = 35 as the cut point)**			
Low (≤35)	4.56 (1.56–13.38)	3.76 (1.17–12.05)	0.03 *
High (>35)	1		
**Practical need** **(using the mean = 13 as the cut point)**			
Low (≤13)	4.64 (1.47–14.70)	5.27 (1.40–19.83)	0.01 *
High (>13)	1		
**Palliative knowledge** (continuous variable)	1.06 (0.88–1.28)	1.28 (1.00–1.63)	0.05 *

Note: 1. Multiple logistic regression using the backward stepwise method with Hosmer and Lemeshow test with Chi-square = 5.54, df 8, *p*-value = 0.70. 2. * = significant predictor.

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
