# Peer review of "Factors Influencing the Willingness of Palliative Care Utilization among the Older Population with Active Cancers: A Case Study in Mandalay, Myanmar"

_ijerph, 2021, doi:10.3390/ijerph18157887_

Round 1

Reviewer 1 Report

The cross-sectional survey of Aye Tinzar Myint and colleagues aimed to examine the factors that might correlate with cancer patients’ willingness to utilize palliative care services in three private oncology clinics in Mandalay, Myanmar. Due to restrictions during the covid-19 pandemic, only 75% of the originally planned number of >50 years old cancer patients (n = 196) were recruited for face-to-face interviews in order to collect data on socio-demographic characteristics including social networks (part I), medical history (part II), knowledge of palliative care (part III), subjective psychosocial, spiritual and practical needs related to palliative care (part IV), and willingness to utilize palliative care services (part V).

The study found that approximately 85% out of 141 older cancer patients were willing to receive palliative care services. The significant predictors of willingness to utilize palliative care services included higher income level, better palliative care knowledge, more need for spiritual and psychosocial support, and practical support.

In order to fully appreciate the authors' work, however, the paper needs some significant changes and additional information should be given to the reader.

The reviewer's suggestions sorted by manuscript section:

1. Introduction:

Minor point: please be consistent with the use of abbreviations (give full explanation only at first appearance, then use abbreviation: e.g. WHO on page 1 and 2).

Major point: please check language; it is not clear what the authors want to say in this phrase on page 2 ("As recommended by the World Health Organization, palliative care is applicable early in the course of illness, in conjunction with other therapies to make patients with prolonging life, such as chemotherapy or radiation therapy [5,12].").

Also in the discussion there are phrases which are difficult to read/to understand (e.g., on page 8: "A study in Southeast Asia, [43]Indonesia, supported spirituality/religious aspects on palliative care provision. They found a significant element in the Indonesian palliative care service and many different religions among the participants. All participants in palliative service reported being affiliated with particular religions and performing their religious practices daily with much comfort[43]" - what do the authors want to say here?).

Please explain also the precise meaning of the phrase "only 0.7% of utilization at palliative care service" on page 2 (i.e., what is the "100%" basis of this calculation?).

Furthermore, some questions related to the methodology section require additional explanation, maybe also in the introduction.

2. Methodology

Please clarify: the study was conducted in private care facilities in Mandalay. It is non known whether palliative care is always private and has to be payed for by the user, or if there is any form of health insurance available (for the general population or for specific groups), that may cover such a service. Furthermore, it would be interesting to know, whether palliative care services are offered in rural areas as well as in bigger cities.

In addition, some information on pricing of palliative care services should be given: how does cost relate to median household income? It might also be helpful for the reader to relate data on houshold income in Kyats to exchange rates at the time of study (e.g., Renminbi Yuan, Yen, US-$, or €).

Table 2.

Please explain, how the cut-off value of 700,000 Kyats per year and more indicating high family income was defined and how it is related to median household income in [urban and rural] Myanmar.

Furthermore, it is not clear how the cut-off value of 40 for social network scores was defined and what such a score does imply. At least some explanation is needed, simply citing the original paper describing the methodology is not enough.

In Table 2 and in the text the authors should give some information on the type of tumor and organ involvement in addition to tumor staging (please clarify, which staging system was used - WHO? year?). If there are no data available, the possibility of confounding the results of the study by differences between tumor types and organ involvement should be discussed in section 4.

In section 2.3 Data Analysis at least some information should be given on the definition of boundaries defining some of the distinctions shown in Tabele 4 and 5 (e.g., what do "social network" high or low, "Spiritual and psychosocial needs" high or low, "Practical need" high or low mean?). Similarly, some explanation should be given regarding the use of comparators used in logistic regression analysis (Tab. 5).

Section 3 Results and section 4 Discussion: see points related to section 1 Introduction and section 2 Methodology. Improvements in the first 2 sections of the mnanuscript may help the reader to understand the results of the study and the discussion more easily and appropriately.

Author Response

Dear Reviewer,

Thank you very much for your valuable comments, we already fix the manuscript as you recommended, please consider them as the red letter in the attached revision. 

Warmest Respectfully,

The authors

P.S. please see the detail of fixing:

Comments

Introduction

Minor point: please be consistent with the use of abbreviations (give full explanation only at first appearance, then use abbreviation: e.g. WHO on page 1 and 2).

Author: Already Fix: WHO as page 1, 2, and 8

Major point: please check language; it is not clear what the authors want to say in this phrase on page 2 ("As recommended by the World Health Organization, palliative care is applicable early in the course of illness, in conjunction with other therapies to make patients with prolonging life, such as chemotherapy or radiation therapy [5,12][1].").

Author: Already rephrase

Also in the discussion there are phrases which are difficult to read/to understand (e.g., on page 8: "A study in Southeast Asia, [43]Indonesia, supported spirituality/religious aspects on palliative care provision. They found a significant element in the Indonesian palliative care service and many different religions among the participants. All participants in palliative service reported being affiliated with particular religions and performing their religious practices daily with much comfort[43]" - what do the authors want to say here?).

Author: Already rephrase

Please explain also the precise meaning of the phrase "only 0.7% of utilization at palliative care service" on page 2 (i.e., what is the "100%" basis of this calculation?).

Author: We decided to remove this confusing phrase and put the reference on the sentence.

 Methodology

Please clarify: the study was conducted in private care facilities in Mandalay. It is non known whether palliative care is always private and has to be payed for by the user, or if there is any form of health insurance available (for the general population or for specific groups), that may cover such a service. Furthermore, it would be interesting to know, whether palliative care services are offered in rural areas as well as in bigger cities.

Author:Palliative care is accessed only at government hospital, but due to difficulty in data conducting at government hospital because of covid 19 pandemic, the researcher has to change to private oncology clinic and study the WILLINGNESS to utilize palliative care services. And health insurance is not available in Myanmar.

In addition, some information on pricing of palliative care services should be given: how does cost relate to median household income? It might also be helpful for the reader to relate data on houshold income in Kyats to exchange rates at the time of study (e.g., Renminbi Yuan, Yen, US-$, or €).

Author: The government hospital is free of charge and palliative care isn't available at private facilities.                               

Table 2.

Please explain, how the cut-off value of 700,000 Kyats per year and more indicating high family income was defined and how it is related to median household income in [urban and rural] Myanmar.

[2]

Author:Using P75  at the cut point as explained in 2.2 and in the table 2

Furthermore, it is not clear how the cut-off value of 40 for social network scores was defined and what such a score does imply. At least some explanation is needed, simply citing the original paper describing the methodology is not enough.

Author: Using Mean cut point as explained in 2.2 and in the table 2

In Table 2 and in the text the authors should give some information on the type of tumor and organ involvement in addition to tumor staging (please clarify, which staging system was used - WHO? year?). If there are no data available, the possibility of confounding the results of the study by differences between tumor types and organ involvement should be discussed in section 4.

Author: We already added the cancer types as confounding factors and re-analyze the data.

In section 2.3 Data Analysis at least some information should be given on the definition of boundaries defining some of the distinctions shown in Tabele 4 and 5 (e.g., what do "social network" high or low, "Spiritual and psychosocial needs" high or low, "Practical need" high or low mean?). Similarly, some explanation should be given regarding the use of comparators used in logistic regression analysis (Tab. 5).

Author: Already fix (red letters)

Section 3 Results and section 4 Discussion: see points related to section 1 Introduction and section 2 Methodology. Improvements in the first 2 sections of the manuscript may help the reader to understand the results of the study and the discussion more easily and appropriately.

Author: We already fix section 2 by adding the confounding factor as cancer types, contributing the results as Table 5 was changed. With this improvement, we can discuss it in conclusion appropriately; please see the change under here (red letter):

This research indicated that approximately 85% of older cancer patients are willing to receive palliative care services. The significant predictors of willingness to utilize palliative care services include place of living, palliative care knowledge,  the need for spiritual and psychosocial support, and practical support. Namely, the patients with more spiritual and psychosocial needs are more likely to be willing to use palliative care 3.76 times than those with a lower need for this support. Regarding practical support, we found that the patients with more need of practical support are more likely to be willing to use palliative care 5.27 times than those with a lower need for support. In addition, patients in rural areas are more likely to be ready to utilize palliative services 3.62 times than those in urban areas. And patients with better knowledge of palliative care are more likely to use palliative care than those with lower palliative knowledge 1.28 times. 

Regarding geographic living, previous research [32,38,39] consistently reported a higher rate of palliative care utilization in patients living in urban than in rural. The results may be possible that people living in rural areas may encounter distance limitation to the facilities and difficulty accessing those services, contributing to the lower utilization rate [32]. However, our result is not the same as the previous research above; we found that patients living in rural areas were more willing to utilize palliative care services than patients in urban areas.  Earlier research from Myanmar revealed that Myanmar older adults in rural areas have unsafe socio-demographic conditions and low healthcare-seeking behavior[40]. Another study supported that Myanmar cancer patients with low socio-economic status (SES) had significantly worse quality of life, perceived poor health care coordination, and were more likely to report severe pain than those with high SES [34]. Therefore, it is reasonable to consider that Myanmar patients living in rural areas may encounter a low quality of care and difficulty accessing health care appropriately. Thus, they may prefer to rely on an alternative medicine associated with traditional spiritual care as palliative practice. With this reason, we strongly believe that the developing community-level palliative care services in Myanmar will be able to facilitate people, especially  in rural areas to have more quality of life and care.

Reviewer 2 Report

The study offers interest as only 20 countries out of 232 countries have integrated palliative care as part of their health care system.

Furthermore, 80% of low- and middle-income countries have a severe shortage of palliative care facilities.

This study indicates that older adults with higher income were more likely to be willing for palliative utilization.

And the same can be said with the older adults with better palliative knowledge and older adults with higher need for practical support and spiritual and psychosocial support.

The interest of these conclusions is relative. Some additional checking is required on English language and expression. For instance, some language edition is required in "developing medicines for the critically ill and the dying is challenging to a political challenge..." (Discussion section).

The same is applicable to "the most need in Spiritual and psychosocial needs is  the item I need spiritual support" (Discussion section). The association between categorized independent variables and the willingness to utilize palliative care requires further explanation.

Author Response

Dear The reviewer:

Thank you very much for your valuable comments on our manuscript. We already fix the revision based on your recommendation. We highlight the fixed ones with yellow letters for you to consider.

Warmest respectfully,

The authors

Round 2

Reviewer 1 Report

The manuscript now reads better than the previous version. However, there are still sentences that might be better understandable after English language editing.